# Regional variations in childbirth interventions and their correlations with adverse outcomes, birthplace and care provider: A nationwide explorative study

Anna E. Seijmonsbergen-Schermers[1]*, Dirkje C. Zondag[1], Marianne Nieuwenhuijze[2], Thomas van den Akker[3], Corine J. Verhoeven[1,4], Caroline C. Geerts[1], François G. Schellevis[5,6], Ank de Jonge[1]

1 Department of Midwifery Science, AVAG, Amsterdam Public Health Research Institute, Amsterdam UMC, Vrije Universiteit Amsterdam, Amsterdam, The Netherlands, 2 Research Centre for Midwifery Science, Zuyd University, Heerlen, The Netherlands, 3 Department of Obstetrics, Leiden University Medical Center, Leiden, The Netherlands, 4 Department of Obstetrics and Gynaecology, Maxima Medical Centre, Veldhoven, The Netherlands, 5 NIVEL (Netherlands Institute for Health Services Research), Utrecht, The Netherlands, 6 Department of General Practice & Elderly Care Medicine, Amsterdam Public Health Research Institute, Amsterdam UMC, Amsterdam, The Netherlands

* a.seijmonsbergen@amsterdammumc.nl

## Abstract

### Background

Variations in childbirth interventions may indicate inappropriate use. Most variation studies are limited by the lack of adjustments for maternal characteristics and do not investigate variations in adverse outcomes. This study aims to explore regional variations in the Netherlands and their correlations with referral rates, birthplace, interventions, and adverse outcomes, adjusted for maternal characteristics.

### Methods

In this nationwide retrospective cohort study, using a national data register, intervention rates were analysed between twelve regions among single childbirths after 37 weeks' gestation in 2010–2013 (n = 614,730). These were adjusted for maternal characteristics using multivariable logistic regression. Primary outcomes were intrapartum referral, birthplace, and interventions used in midwife- and obstetrician-led care. Correlations both between primary outcomes and between adverse outcomes were calculated with Spearman's rank correlations.

### Findings

Intrapartum referral rates varied between 55–68% (nulliparous) and 20–32% (multiparous women), with a negative correlation with receiving midwife-led care at the onset of labour in two-thirds of the regions. Regions with higher referral rates had higher rates of severe postpartum haemorrhages. Rates of home birth varied between 6–16% (nulliparous) and 16–

**Data Availability Statement:** Data cannot be shared publicly because of restrictions of the perinatal register "Perined". Data are available from Perined (contact via info@perined.nl) for researchers who meet the criteria for access to confidential data, and if Perined gives permission. Data can be requested by Perined by filling in the application form "Aanvraagformulier gegevens" that can be found on the website https://www.perined. nl/registratie/faciliteren-onderzoek. In our study, the datasets of the years 2010, 2011, 2012, and 2013 were used, which are available at the Department of Midwifery Science of Amsterdam UMC, location VUmc. The following variables were requested prior to the start of the study: From LVR 1 and 2: perineal damage, maternal age, ethnic background, gravidity, parity, gestational age at time of birth, certainty due date, consultation/ referral to secondary care and reason of referral, birth of child attended by, under responsibility/ supervision of, amniotomy, amniotic fluid colour, time start pushing, fetal position, medication after birth child, blood loss, birth date, birth time, Apgar score, birth weight, planned and actual place of birth, consultation paediatrician, referral paediatrician, other problems child, list particularities, other problems mother. From LVR 2: multiple birth number, referral from primary or secondary care, reason of referral from primary care, attended by own practice, onset of labour, indication induction or prelabour caesarean section, stimulation for lack of labour progression, pain medication, birth assistance, indication for assistance or intrapartum caesarean section, afterbirth period, congenital malformations, paediatric involvement, particularities.

**Funding:** This study was funded by AVAG, Amsterdam Public Health Research Institute, Amsterdam UMC. There was no external funding for this study. The corresponding author had full access to all the data in the study and had final responsibility for the decision to submit for publication.

**Competing interests:** The authors declare that they have no competing interests.

31% (multiparous), and was negatively correlated with episiotomy and postpartum oxytocin rates. Among midwife-led births, episiotomy rates varied between 14–42% (nulliparous) and 3–13% (multiparous) and in obstetrician-led births from 46–67% and 14–28% respectively. Rates of postpartum oxytocin varied between 59–88% (nulliparous) and 50–85% (multiparous) and artificial rupture of membranes between 43–52% and 54–61% respectively. A north-south gradient was visible with regard to birthplace, episiotomy, and oxytocin.

## Conclusions

Our study suggests that attitudes towards interventions vary, independent of maternal characteristics. Care providers and policy makers need to be aware of reducing unwarranted variation in birthplace, episiotomy and the postpartum use of oxytocin. Further research is needed to identify explanations and explore ways to reduce unwarranted intervention rates.

## Introduction

Rates of interventions during childbirth have been studied worldwide and large variations between countries have been reported [1]. Interventions during childbirth can be crucial in order to prevent neonatal and maternal morbidity and mortality [2] and, therefore, underuse of healthcare services can be an important cause of preventable morbidity and mortality. On the other hand, use without a medical indication may cause avoidable harm, given the risk of adverse effects related to interventions [3–5].

Worldwide, rates of most interventions and referrals during childbirth have increased [1, 6, 7], episiotomy being the exception [1, 8]. The rate of home births varies worldwide and is low in most high-income countries. For instance, in 2017, the rate of home births was 1% in the USA [9], 0.3% in Australia [10], and 2% in England and Wales [11]. The rate of referrals depends on the maternity care system in a country. Alliman et al. (2016) showed a range of intrapartum referral rates from birth centres to hospitals of between 12% and 37% [12], and Blix et al. (2014) a referral rate range from home to hospital of between 10% and 32% [13]. Episiotomy rates vary largely, from 5% in Denmark, to 75% in Cyprus [14].

In the Netherlands, a variation in intrapartum referral rates among midwifery practices has been shown of between 10% and 64% [15]. Since the late twentieth century, referrals during pregnancy and labour have increased continuously. Where in 1999 more than 60% of women received midwife-led care at the onset of labour, this number has decreased to 51% in 2015 [16, 17]. The rate of home births has historically been high in the Netherlands. However, the rate of home births declined from 23% in 2000 to 13% in 2015 [16, 17]. Episiotomy rates declined from 23% in 2005 to 13% in 2015 in the Netherlands.

Variations in the use of interventions between countries may be explained by differences in maternal and perinatal characteristics and perinatal healthcare systems [18]. For example, parity, maternal age, ethnicity, and birthweight are associated with the use of episiotomy [19]. However, since these factors are likely to be more similar within a specific country, less variation may be expected within a country than between countries [20]. If variations persist after adjustment for maternal characteristics, then this may indicate that variations are unwarranted and it may indicate inappropriate use [20, 21]. While on one hand characteristics can be associated with interventions, on the other hand interventions can be associated with improved or worsened neonatal and maternal outcomes [2–5]. Therefore, intervention rates should be

investigated in view of adverse outcomes. Similar neonatal and maternal outcomes between regions together with variations in interventions, is another indicator of unwarranted variation [20, 21].

A first step in addressing possible inappropriate use of interventions is to examine regional variations of intrapartum interventions within a country [20]. Regional variation would not be expected in a relatively small country such as the Netherlands without regional differences in the maternity healthcare system. In the Dutch maternity care system, low-risk women start antenatal care in midwife-led primary care. Midwives refer women to obstetrician-led care when risks of adverse outcomes increase or complications arise. Interventions such as episiotomy, artificial rupture of membranes (AROM), and postpartum administration of oxytocin are used in both midwife-led and obstetrician-led care settings [22]. Box 1 citesthe description of the maternity care system in the Netherlands from our previous publication on regional variations in the Netherlands [23].

The decreased rate of midwife-led births in the Netherlands corresponded with an increased use of interventions on the national level [15, 28]. A previous article described regional variations in rates of induction and augmentation of labour, pain medication, caesarean section, and involvement of paediatrician in the first 24 h after birth [23]. Little information is available as to how regional variations in rates of referral, place of birth, and interventions during childbirth, relate to each other, nor how they might relate to adverse neonatal and maternal outcomes. Knowledge on these correlation will give insight into underlying processes of variations in childbirth interventions, place of birth, and referral, and will help care providers and policy makers to know which variation is large and likely unwarranted and should therefore be the focus of changes in practices and policies with the ultimate aim to improve the quality of maternity care. This article focuses on rates of referral, place of birth, and interventions that are used in both primary midwife-led, and secondary obstetrician-led

## Box 1. The maternity care system in the Netherlands [23]

In the Netherlands, there are no regional differences in the maternity healthcare system. Low-risk women start antenatal care in midwife-led primary care. These women are cared for by independent midwives who attend home births, low-risk hospital births, and births in alongside and free-standing birth centres. The Dutch Birth Centre Study showed that health outcomes, experiences, and costs for low-risk women are similar for planned birth in a birth centre and planned birth in a hospital, both supervised by a primary care midwife [24, 25]. Midwives refer women to obstetrician-led care when risks of adverse outcomes increase or complications arise. Criteria for referral from midwife-led to obstetrician-led care have been laid out in the obstetric indication list of 2003. Interventions in childbirth such as induction and augmentation of labour, pain medication, instrumental birth, and CS, are only available in an obstetrician-led care setting [22, 26]. These intrapartum interventions may be used for women in midwife-led care at the onset of labour after referral to obstetrician-led care. Interventions such as episiotomy, artificial rupture of membranes (AROM), and postpartum administration of oxytocin are used in both midwife-led and obstetrician-led care settings [22]. The Steering Committee 'Pregnancy and birth' recommended in 2009 more integration in maternity care between midwife-led and obstetrician-led care, which led to several regional initiatives to change the organisation of care, but there was not one uniform, national model [27].

care. The first aim of this study was to explore which regional variations in intrapartum rates of referral, place of birth, and use of intrapartum interventions, exist for women who gave birth in the Netherlands between 2010 and 2013. Secondly, we aimed to investigate how these variations are correlated to each other. Thirdly, we examined the association between variations, and adverse neonatal and maternal outcomes, adjusted for maternal characteristics.

## Methods

### Data collection

The methods of this explorative study have been described previously in more detail [23]. For this nationwide study, we used data on single births after 37 weeks of gestation. We focused on single births because multiple pregnancies are associated with much higher risks of adverse outcomes and therefore medical interventions are often justified. These data originated from the national register, "Perined", covering the years 2010 up to 2013 and including 98% of all births in the Netherlands after 24 weeks of gestation [29]. Patient records were excluded when data were missing on: postal codes or parity; or when data were missing from the midwifery database of women that received both midwife- and obstetrician-led care. The pitfalls in the use of data based on the national register have been described in a recently published article [30].

### Selection of variables

We used the twelve Dutch administrative provinces as regions. A record of a birth was allocated to a region on the basis of the mother's residential postal code. All women, in all regions, have access to all types of birth settings and therefore, these regions were comparable.

**Care processes.**   The following primary outcome variables concerning maternity care processes were examined: the number of women receiving midwife-led care at the onset of their pregnancy, at the onset of labour, and at the time of birth; intrapartum referral to obstetrician-led care, and the planned, and actual place of birth (home, hospital and birth centre midwife-led, hospital obstetrician-led). The onset of labour was defined as the onset of active uterine contractions or rupture of membranes. Intrapartum referral to an obstetrician-led care setting was defined as a referral after the onset of labour and before birth. The planned place of birth was defined as that at the onset of labour. Therefore, women who were referred, during pregnancy and before labour, to obstetrician-led care could not have a birth planned in midwife-led care.

**Interventions.**   The following primary outcome variables concerning interventions were examined: the rates of episiotomy in vaginal births; AROM, and postpartum administration of oxytocin. Data about AROM and postpartum administration of oxytocin were only available in the midwifery part of the perinatal database and were therefore not described for the obstetrician-led care group.

**Adverse outcomes.**   Secondary neonatal and maternal outcomes were: antepartum and intrapartum stillbirth; neonatal mortality up to seven days; Apgar score below seven at five minutes; third or fourth degree perineal tear among vaginal births; and postpartum haemorrhages (PPH) of more than 1,000 ml.

### Data analysis

The differences between primary outcomes per region were analysed initially as a whole and then in subgroups of women in midwife-led or in obstetrician-led care, at the onset of labour and at the time of birth. Crude rates were given separately for nulliparous and multiparous

women since rates of interventions are not comparable for these groups. The analyses were conducted on the level of the women, with the region as independent variable, and the primary or secondary outcome as dependent variable. In the logistic regression analyses, the overall rate of the outcome we investigated, weighted for the number of women per region, was considered as a reference. Univariable analyses were conducted in order to gain insight into the variations in the rates of primary and secondary outcomes among the twelve regions. These were followed by multivariable logistic regression analyses with adjustments for: maternal age (<40 years, 40 years or older); ethnic background (Dutch, non-Dutch); socioeconomic position (low, medium or high)—based on postal code and education, employment and level of income; and the degree of urbanisation (rural, intermediate or urban)—based on women's residential postal code (Statistics Netherlands). A confidence interval of 99% was chosen to account for multiple testing within a large dataset. The aim of these analyses was to explore differences between the regions, which are not explained by maternal characteristics, but may be explained by variations between care professionals and/or care settings such as midwifery practices and hospitals. Therefore, we did not perform multilevel analyses.

Figures with maps of the regions and boxplots with adjusted odds ratios (ORs) and confidence intervals were used to visualise the results. To test whether variables with a significant regional variation correlated with each other, a Spearman's rank correlation coefficient was calculated. Correlations in the 12 regions with rho $\geq 0.57$ or $\leq - 0.57$ corresponded with a p-value of 0.05, and a correlation of rho $\geq 0.60$ or $\leq - 0.60$ was considered strong [31]. Statistical analyses were performed using SPSS Statistics 22 (SPSS Inc, Chicago, IL, USA).

## Results

### Overall variation

The total number of women in this study was 614,730. The number of women in midwife- or obstetrician-led care at the onset of labour and at the time of birth, for nulliparous and multiparous women separately, are presented in Fig 1. Maternal characteristics have previously been published [23]. Fig 2 shows that the largest regional variations for the use of episiotomy and postpartum administration of oxytocin, were found in women who received midwife-led care at the time of birth. Adjustments for maternal age, ethnic background, socioeconomic position and the degree of urbanisation did not lead to substantial changes in regional variation. Therefore, we have shown the adjusted ORs only in the supplementary tables. Lower rates of episiotomy and postpartum administration of oxytocin were found in regions where home births were more common (rho = - 0.60 and—0.79, respectively; S2 Table).

### Variations in referral and place of birth

Table 1 shows the rates of the primary outcomes for each region. In nulliparous women, variations in regions were found of between 84% and 93% for being in midwife-led care at the onset of pregnancy, of between 47% and 61% at the onset of labour, and of between 17% and 26% at the time of birth. These rates were similar for multiparous women, with the exception that there were higher levels for women receiving midwife-led care at the time of birth, with a variation of between 33% and 43%. Correlations are shown in S2 and S3 Tables, which are available in the supplementary material on the web. In regions with more women in midwife-led care at the onset of labour, there were more births in midwife-led care (rho = 0.84) and more home births (rho = 0.69; S2 Table).

Adjusted ORs of variations in the rates of primary outcomes can be found in S1 Table. Adverse neonatal and maternal outcomes have been described in a previous article [23]. A visualisation of the adjusted ORs is provided in Figs 2–7. Fig 3 shows that in eight regions the

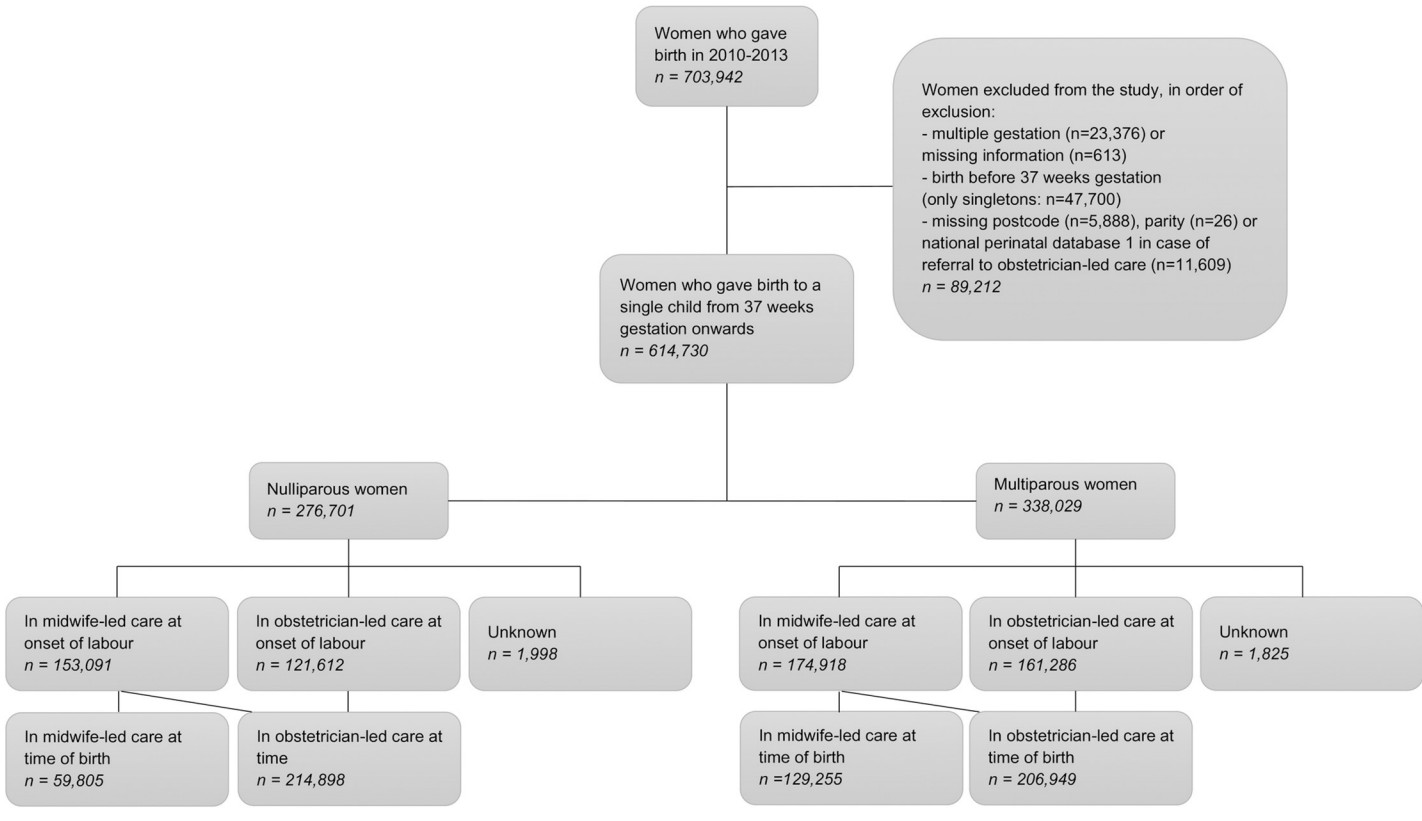

**Fig 1. Study population.**

number of women in midwife-led care at the onset of labour was negatively correlated with intrapartum referral rates to obstetrician-led care, but the overall correlation was not significant (rho = - 0.40; S2 Table). Rates of intrapartum referrals from midwife-led to obstetrician-led care varied from between 55% to 68% for nulliparous and from between 20% to 32% for multiparous women (Table 1). The correlations are shown in S3 Table. In regions where referral rates were higher, rates of PPH were higher as well (rho = 0.74; S3 Table). Rates of planned home births varied from between 7% to 30% among nulliparous and from between 11% to 32% among multiparous women. For actual home birth, rates varied from between 6% to 16% among nulliparous and from between 16% to 31% among multiparous women (Table 1). Planned and actual home birth were strongly correlated with each other (rho = 0.98; S2 Table).

## Variations in childbirth interventions

There was a considerable north-south divide with regard to the place of birth, episiotomy and postpartum administration of oxytocin. Rates of home births were higher in the north of the country than in the south (Fig 2), and were negatively correlated with episiotomy (rho = - 0.60; S2 Table).

Rates of postpartum administration of oxytocin among women in midwife-led care at the time of birth varied from between 59% to 88% among nulliparous women and from between 50% to 85% among multiparous women (Table 1). Overall, among women in midwife-led care, the correlation of actual home birth and postpartum administration of oxytocin was rho = - 0.79 (Fig 4). In regions where midwives more frequently administer oxytocin postpartum, overall rates of PPH were not lower (rho = 0.08; S3 Table).

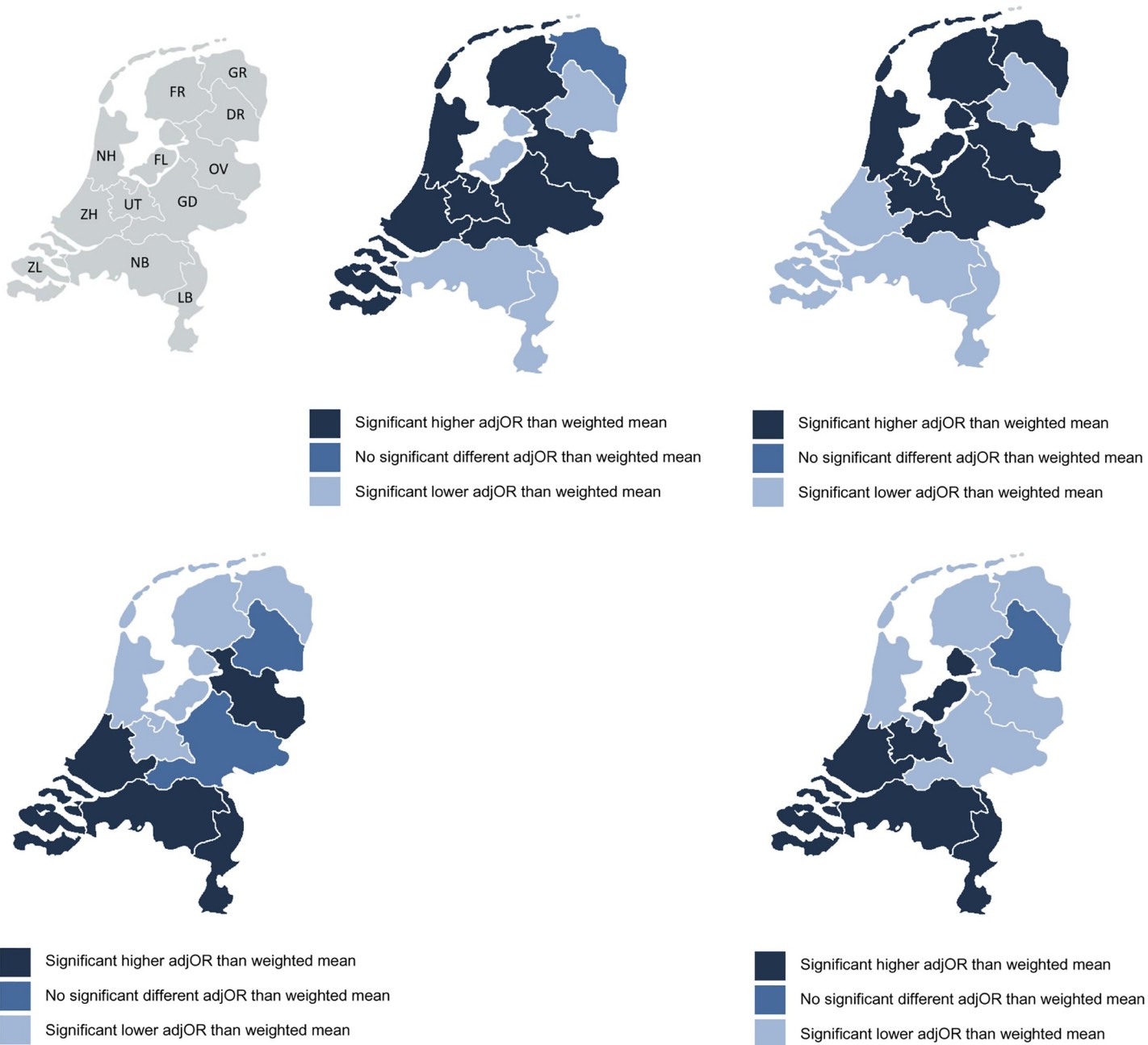

**Fig 2. A.** Interregional variation in adjOR* of number of women in midwife-led care at time of birth. * Adjusted for parity, maternal age, ethnic background, socioeconomic position and urbanisation. **B.** Interregional variation in adjOR* of actual home births. * Adjusted for parity, maternal age, ethnic background, socioeconomic position and urbanisation. **C.** Interregional variation in adjOR* of episiotomy. * Adjusted for parity, maternal age, ethnic background, socioeconomic position and urbanisation. **D.** Interregional variation in adjOR* oxytocin postpartum among women in midwife-led care. * Adjusted for parity, maternal age, ethnic background, socioeconomic position and urbanisation.

For episiotomy, a variation of between 39% and 60% was found in nulliparous women, and of between 10% and 21% among multiparous women. Variation was greatest for women receiving midwife-led care at the time of birth, with rates varying from between 14% to 42% among nulliparous women and from between 3% to 13% among multiparous women. Among women receiving obstetrician-led care, rates varied from between 46% to 67% among

**Table 1. Primary outcomes: Process of care and intervention rates by region.**

*All nulliparous women*

| | Total | GR | FR | DR | OV | FL | GD | UT | NH | ZH | ZL | NB | LB |
|---|---|---|---|---|---|---|---|---|---|---|---|---|---|
| **Total *n*** | **276,701** | 8,901 | 9,564 | 6,785 | 18,051 | 7,233 | 30,961 | 23,662 | 49,582 | 63,785 | 4,853 | 38,544 | 14,780 |
| **Women in midwife-led care, %** | | | | | | | | | | | | | |
| At onset of pregnancy | **88.5** | 90.4 | 92.5 | 90.6 | 90.0 | 90.5 | 91.2 | 92.4 | 89.9 | 83.6 | 89.5 | 89.1 | 83.7 |
| At onset of labour | **55.7** | 54.4 | 59.8 | 52.5 | 57.4 | 55.2 | 60.2 | 60.7 | 59.8 | 52.0 | 55.7 | 52.6 | 46.7 |
| At time of birth | **21.8** | 21.7 | 25.6 | 19.1 | 24.8 | 17.8 | 23.8 | 21.9 | 23.0 | 22.6 | 25.1 | 17.2 | 17.4 |
| **Planned place of birth, %** | | | | | | | | | | | | | |
| Home | **16.7** | 17.8 | 30.1 | 19.1 | 26.2 | 16.7 | 24.5 | 16.3 | 15.1 | 12.4 | 6.8 | 15.0 | 10.5 |
| Hospital/birth centre midwife-led | **30.0** | 33.5 | 26.1 | 30.5 | 25.2 | 34.1 | 28.4 | 35.2 | 33.9 | 29.9 | 30.5 | 26.4 | 25.7 |
| Hospital obstetrician-led | **44.2** | 45.6 | 40.2 | 47.5 | 42.6 | 44.8 | 39.7 | 39.3 | 39.9 | 47.8 | 44.2 | 47.3 | 53.2 |
| Other/unknown | **9.1** | 3.1 | 3.6 | 2.8 | 5.9 | 4.4 | 7.4 | 9.3 | 11.1 | 9.9 | 18.5 | 11.3 | 10.5 |
| **Actual place of birth*, %** | | | | | | | | | | | | | |
| Home | **9.8** | 11.3 | 15.8 | 10.3 | 14.0 | 9.3 | 14.4 | 9.6 | 10.1 | 7.4 | 5.5 | 7.7 | 7.3 |
| Hospital/birth centre midwife-led | **10.9** | 10.0 | 9.5 | 8.7 | 10.3 | 8.4 | 8.9 | 11.5 | 11.6 | 13.5 | 18.9 | 8.6 | 9.7 |
| Hospital obstetrician-led | **79.2** | 78.6 | 74.7 | 81.1 | 75.7 | 82.3 | 76.7 | 78.9 | 78.3 | 79.1 | 75.6 | 83.7 | 83.1 |
| **Episiotomy in vaginal births, %** | **47.4** | 42.1 | 44.4 | 49.2 | 52.0 | 39.1 | 47.4 | 43.2 | 38.9 | 49.4 | 59.9 | 54.7 | 55.7 |

*All multiparous women*

| | Total | GR | FR | DR | OV | FL | GD | UT | NH | ZH | ZL | NB | LB |
|---|---|---|---|---|---|---|---|---|---|---|---|---|---|
| **Total *n*** | **338,029** | 10,540 | 13,004 | 9,090 | 24,818 | 10,228 | 40,325 | 29,231 | 56,366 | 75,788 | 6,474 | 45,643 | 16,522 |
| **Women in midwife-led care, %** | | | | | | | | | | | | | |
| At onset of pregnancy | **84.1** | 84.8 | 88.6 | 83.6 | 86.8 | 83.3 | 87.0 | 88.1 | 84.9 | 79.7 | 84.4 | 84.2 | 79.2 |
| At onset of labour | **52.0** | 49.3 | 55.5 | 47.8 | 55.2 | 48.6 | 56.9 | 56.0 | 54.5 | 49.5 | 52.3 | 47.9 | 46.1 |
| At time of birth | **38.4** | 37.9 | 42.7 | 34.9 | 42.6 | 33.5 | 43.0 | 40.7 | 39.9 | 37.2 | 41.9 | 32.8 | 33.9 |
| **Planned place of birth, %** | | | | | | | | | | | | | |
| Home | **20.5** | 21.1 | 32.0 | 22.1 | 29.4 | 19.4 | 28.6 | 20.4 | 19.3 | 16.5 | 10.8 | 17.4 | 13.5 |
| Hospital/birth centre midwife-led | **23.7** | 24.9 | 19.4 | 23.6 | 19.6 | 24.0 | 21.9 | 27.9 | 26.5 | 24.1 | 26.0 | 22.2 | 21.4 |
| Hospital obstetrician-led | **47.9** | 50.7 | 44.5 | 52.2 | 44.8 | 51.4 | 43.1 | 44.0 | 45.3 | 50.3 | 47.7 | 52.0 | 53.8 |
| Other/unknown | **7.9** | 3.3 | 4.1 | 2.1 | 6.2 | 5.2 | 6.4 | 7.7 | 8.9 | 9.1 | 15.5 | 8.3 | 11.3 |
| **Actual place of birth*, %** | | | | | | | | | | | | | |
| Home | **21.4** | 22.7 | 30.5 | 20.6 | 28.8 | 21.0 | 29.3 | 21.6 | 20.7 | 17.4 | 15.9 | 17.3 | 17.3 |
| Hospital/birth centre midwife-led | **16.0** | 14.9 | 11.7 | 14.2 | 13.3 | 12.3 | 13.2 | 18.2 | 17.6 | 18.0 | 25.5 | 14.2 | 16.1 |
| Hospital obstetrician-led | **62.7** | 62.4 | 57.8 | 65.2 | 57.9 | 66.7 | 57.5 | 60.2 | 61.7 | 64.6 | 58.6 | 68.5 | 66.6 |
| **Episiotomy in vaginal births, %** | **14.8** | 11.9 | 13.4 | 15.1 | 16.6 | 9.9 | 15.3 | 12.4 | 10.4 | 14.8 | 20.2 | 19.8 | 20.6 |

*Midwife-led care at onset of labour*

| | Total | GR | FR | DR | OV | FL | GD | UT | NH | ZH | ZL | NB | LB |
|---|---|---|---|---|---|---|---|---|---|---|---|---|---|
| **Total *n*** | **328,009** | 10,013 | 12,901 | 7,884 | 23,922 | 8,942 | 41,401 | 30,601 | 59,750 | 70,023 | 6,068 | 42,040 | 14,464 |
| **Intrapartum transfer to obstetrician-led care, %** | | | | | | | | | | | | | |
| *nulliparous* | **60.9** | 60.2 | 57.1 | 63.5 | 56.8 | 67.7 | 60.5 | 63.9 | 61.5 | 56.6 | 54.9 | 67.2 | 62.8 |
| *multiparous* | **26.1** | 23.1 | 23.0 | 26.9 | 22.8 | 31.1 | 24.4 | 27.3 | 26.8 | 24.8 | 19.7 | 31.6 | 26.5 |
| **AROM, %** | | | | | | | | | | | | | |
| *nulliparous* | **46.9** | 45.3 | 49.5 | 47.9 | 47.3 | 51.6 | 46.8 | 44.4 | 46.9 | 48.1 | 50.0 | 46.1 | 43.3 |
| *multiparous* | **57.3** | 53.9 | 56.3 | 60.1 | 59.1 | 58.2 | 56.8 | 55.3 | 55.4 | 58.9 | 61.0 | 58.7 | 55.1 |

*Midwife-led care at time of birth*

| | Total | GR | FR | DR | OV | FL | GD | UT | NH | ZH | ZL | NB | LB |
|---|---|---|---|---|---|---|---|---|---|---|---|---|---|
| **Total *n*** | **189,060** | 5,912 | 7,985 | 4,461 | 14,962 | 4,706 | 24,600 | 17,014 | 33,578 | 42,223 | 3,922 | 21,557 | 8,140 |
| **Postpartum administration of oxytocin, %** | | | | | | | | | | | | | |
| *nulliparous* | **76.0** | 59.0 | 71.6 | 74.1 | 69.9 | 76.7 | 68.8 | 80.3 | 73.2 | 80.7 | 88.2 | 81.4 | 85.3 |

*(Continued)*

**Table 1.** (Continued)

| | | | | | | | | | | | | |
|---|---|---|---|---|---|---|---|---|---|---|---|---|
| *multiparous* | **69.1** | 50.3 | 61.6 | 70.1 | 60.0 | 72.9 | 61.8 | 72.4 | 66.7 | 73.5 | 84.6 | 77.0 | 80.3 |
| **Episiotomy, %** | | | | | | | | | | | | |
| *nulliparous* | **23.4** | 17.0 | 19.4 | 25.9 | 32.9 | 13.8 | 23.4 | 19.1 | 14.4 | 26.2 | 41.7 | 28.1 | 31.6 |
| *multiparous* | **7.0** | 4.8 | 5.7 | 7.2 | 9.5 | 3.4 | 7.3 | 5.3 | 4.0 | 6.8 | 12.5 | 10.5 | 9.7 |
| ***Obstetrician-led care at time of birth*** | | | | | | | | | | | | |
| | Total | GR | FR | DR | OV | FL | GD | UT | NH | ZH | ZL | NB | LB |
| **Total** *n* | **421,847** | 13,477 | 14,512 | 11,375 | 27,678 | 12,712 | 46,372 | 35,665 | 71,249 | 96,023 | 7,368 | 62,386 | 23,030 |
| **Episiotomy in vaginal births, %** | | | | | | | | | | | | |
| *nulliparous* | **55.7** | 50.8 | 55.4 | 56.0 | 59.7 | 45.7 | 56.5 | 51.4 | 48.2 | 57.3 | 67.4 | 61.6 | 61.9 |
| *multiparous* | **20.9** | 17.3 | 20.6 | 20.4 | 23.3 | 13.9 | 22.6 | 18.5 | 15.9 | 20.4 | 27.2 | 25.5 | 27.8 |

GR = Groningen; FR = Friesland; DR = Drenthe; OV = Overijssel; FL = Flevoland; GD = Gelderland; UT = Utrecht; NH = Noord-Holland; ZH = Zuid-Holland; NB = Noord-Brabant; LB = Limburg. These regions are the twelve provinces in the Netherlands.

Percentage of missing data: 0.0% for women in midwife-led care at onset of pregnancy, 0.6% for women in midwife-led care at onset of labour, 0.6% for women in midwife-led care at time of birth, 0.4% for planned place of birth, 0.6% for actual place of birth, 0.0% for transfer to obstetrician-led care during labour, 3.8% for artificial rupture of membranes, 4.2% for episiotomy, 3.7% for oxytocin postpartum.

* Due to some missing values in the actual place of birth variable, the rates of actual home birth and midwife-led birth in a hospital or birth centre do not add up to the total rate of women in midwife-led care at time of birth.

nulliparous women and from between 14% to 28% among multiparous (Table 1). There was a strong correlation between episiotomy rates in midwife-led and in obstetrician-led care settings within the same region (rho = 0.96, Fig 5). The correlation between the adjusted ORs of actual home birth and episiotomy was rho = - 0.60 (S2 Table). We did not find a correlation between the adjusted ORs of episiotomy and third or fourth degree perineal tear in vaginal births (rho = - 0.20; S3 Table and Fig 6A and 6B).

Least variation was found for AROM among women receiving midwife-led care at the onset of labour (Fig 7). Here rates varied from between 43% to 52% among nulliparous women and from between 54% to 61% among multiparous (Table 1). Except for the positive correlation between intrapartum referral and PPH (rho = 0.74), correlations with other adverse outcomes were not statistically significant.

## Discussion

This study is a first step towards understanding the appropriate use of interventions in childbirth, as was recommended in a Lancet series on Midwifery [2]. Most variation was found in the use of episiotomy and postpartum administration of oxytocin in women receiving midwife-led care at the time of birth. These intervention can be applied in both home and hospital births, but lower rates were found in regions with more home births. Although there was a correlation between episiotomy use in midwife-led care and episiotomy use in obstetrician-led care, the variation between regions in episiotomy rates was greater for women receiving midwife-led care. A finding we believe to be significant was the negative correlation that was found in two-thirds of the regions between the number of women in midwife-led care at the onset of labour and intrapartum referral rates. The only correlation that we found with adverse neonatal and maternal outcomes was between intrapartum referral and PPH; there were higher rates of PPH in regions with more intrapartum referrals, which did not include referrals after birth. Significantly, in our study, no correlation was found between the regional adjusted ORs for episiotomy, and third or fourth degree rupture.

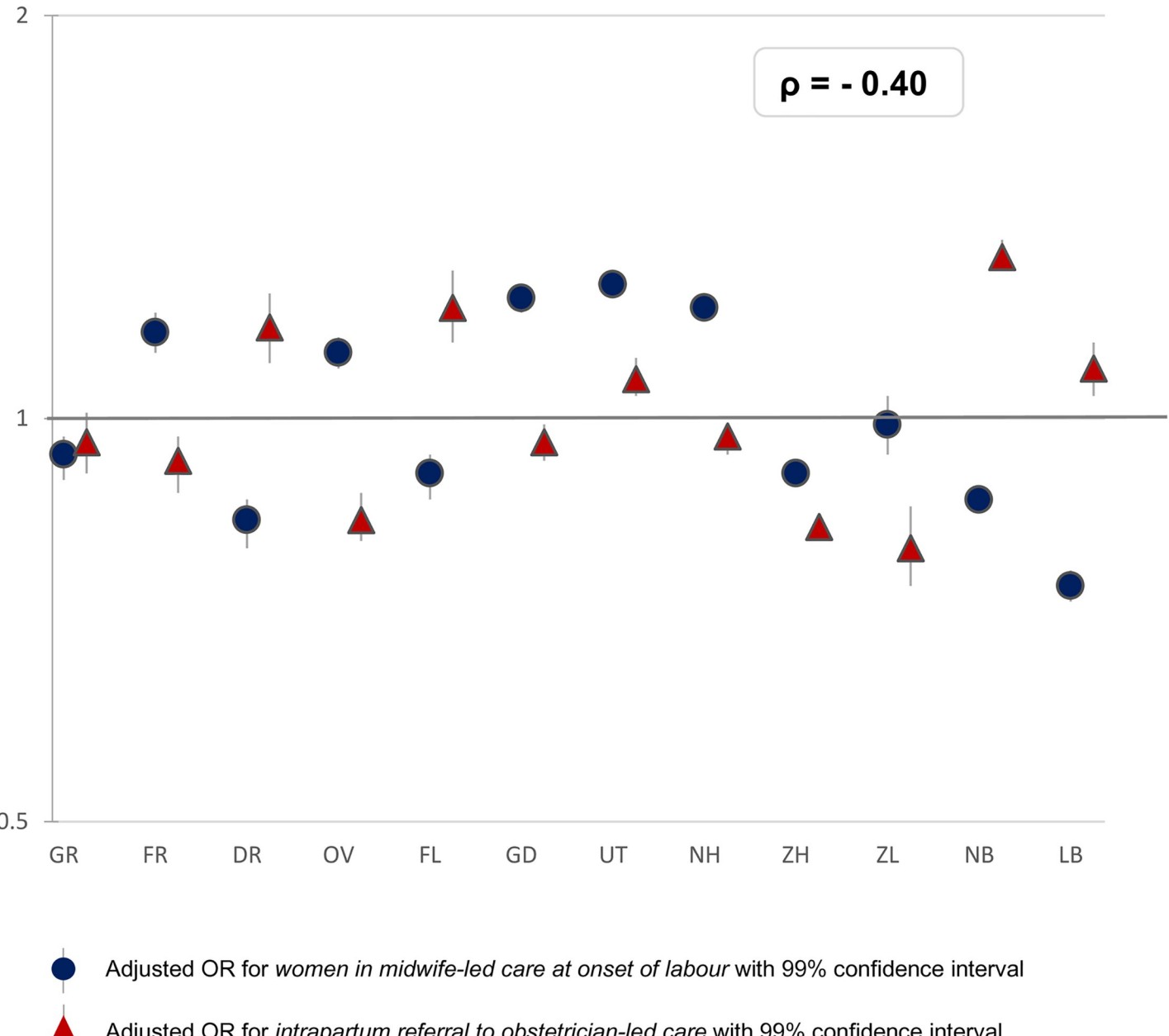

**Fig 3. Interregional variation in midwife-led care at onset of labour and intrapartum referral* to obstetrician-led care.** * Referral to obstetrician-led care during labour: Only women in midwife-led care at the onset of labour are shown. The reference category (OR of 1.0) is the weighted overall rate of the country.

## Limitations and strengths

A limitation of this study is the absence, or low quality of other relevant data in the register such as the maternal body mass index, congenital disorders, and history of obstetric complications. However, adjustments for maternal age, ethnic background, socioeconomic position and degree of urbanisation did not lead to considerable changes in ORs. Therefore, it is unlikely that regional variation in underlying morbidity, including variation in indications for interventions, would explain entirely the variations observed. Reporting bias is an issue for

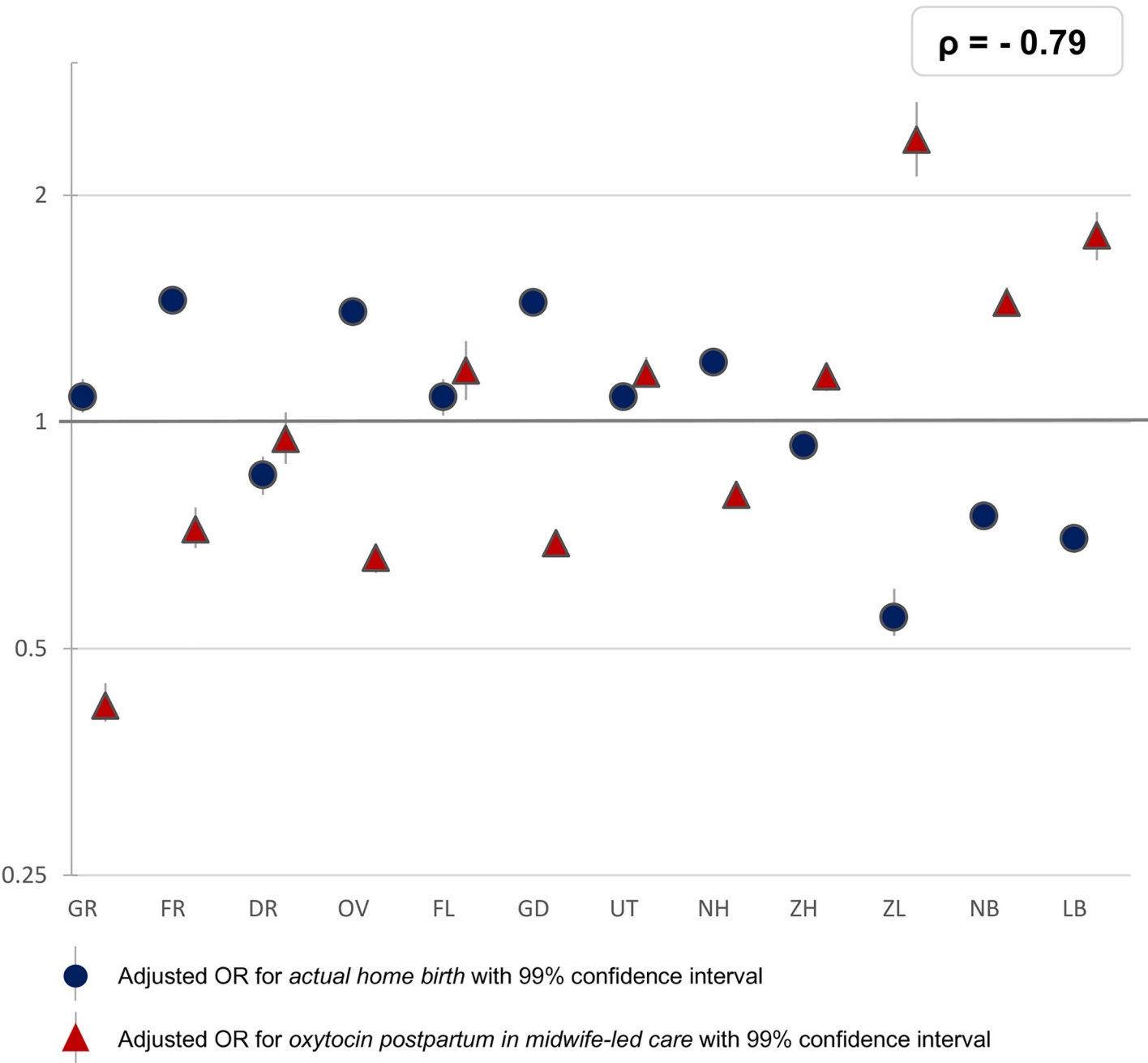

**Fig 4. Interregional variation in actual home birth and oxytocin postpartum in midwife-led care.** The reference category (OR of 1.0) is the weighted overall rate of the country.

datasets based on national registers [30] but we have no indication that misclassifications are different across regions. The number of missing values was very low; there were no outcome variables with more than 1.5% of values missing, and no characteristic variables with more than 2.5% of values missing.

Additionally, the correlation coefficients calculated are only a crude indicator of the relevant and significant correlations between variables, since it was not possible to account

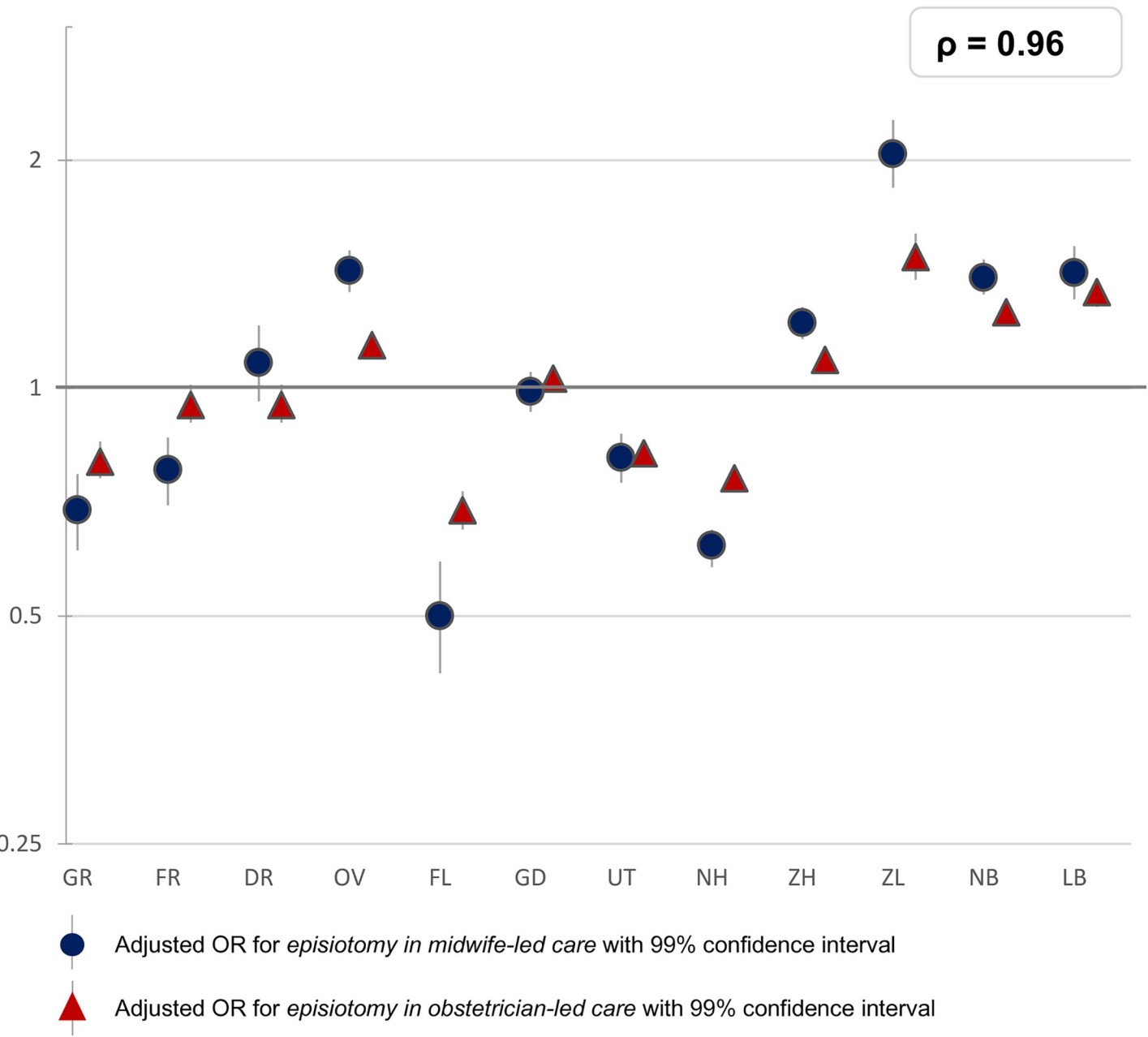

**Fig 5. Interregional variation in episiotomy in vaginal births at the time of birth.** The reference category (OR of 1.0) is the weighted overall rate of the country.

for confidence intervals by calculating Spearman's correlation coefficients of adjusted ORs.

A major strength of this study was access to data of the total population of women in the Netherlands who gave birth in the four-year study period. Furthermore, we were able to differentiate between outcomes for women receiving midwife-led and obstetrician-led care, at the onset of labour and at the time of birth. In this manner, the distinction could be made between groups of women at a low and those at a higher risk of complications. This makes confounding by indication less likely. While not all twelve regions have a university

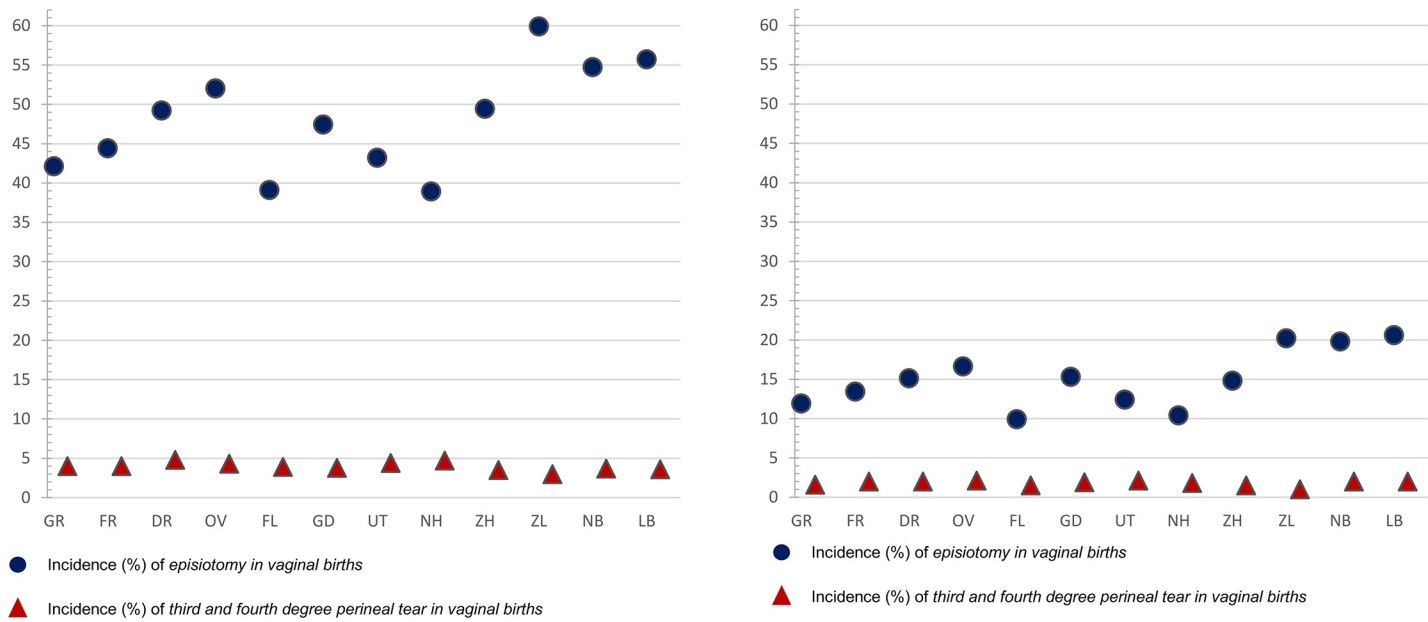

**Fig 6. A.** Crude incidences of episiotomy and 3rd/4th degree perineal tear for nulliparous women with vaginal births. Correlation between adjusted ORs of episiotomy and of 3rd/4th degree perineal tear among all women is rho = - 0.20. **B.** Crude incidences of episiotomy and $3^{rd}/4^{th}$ degree perineal tear for multiparous women with vaginal births. Correlation between adjusted ORs of episiotomy and of 3rd/4th degree perineal tear among all women is rho = - 0.20.

hospital, the analyses were based on the women's residential postal code instead of the postal code of the place of birth, thus the presence or absence of a university hospital in a region will have had a limited influence on the results. However, other confounders, such as distance to a hospital, may have had an influence upon the outcomes and may explain, in part, some variations. Correlations which were investigated in this study cannot be interpreted as causal relationships. Further research is needed to examine whether causality exists for the associations found.

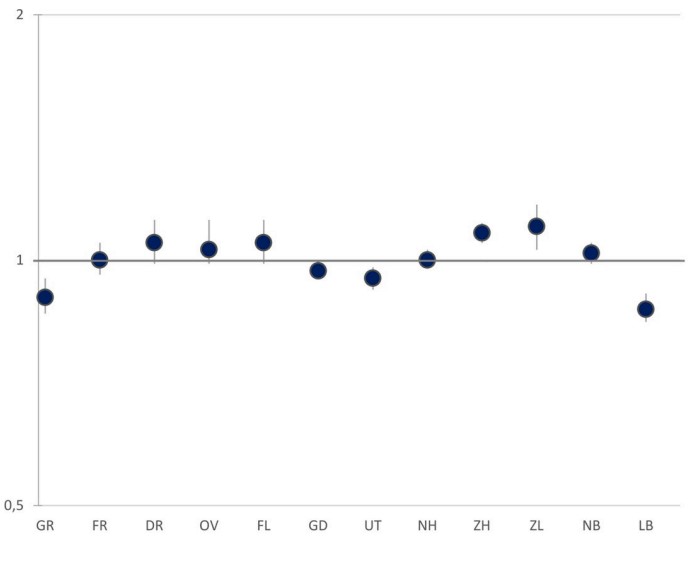

**Fig 7. Interregional variation in artificial rupture of membranes in midwife-led care at onset of labour.**

## Interpretation and further research

**Interpretation in view of previous literature.** Previous studies on regional variations in other countries, showed a variety in intervention rates across studies. When focusing on the subjects of our study, we found lower rates and less variation in rates of AROM, and, similar to our results, large variations in episiotomy rates. Studies were found on regional variations in episiotomy and AROM in Brazil, Ireland, Canada, and France, but there is a lack of literature on regional variations in care processes. A Brazilian study, including 23,940 women, found rates of AROM varying from between 32% to 48%, and episiotomy rates varying from 49% to 69% across five regions, although variation in episiotomy rates was only significantly different in the region with the highest incidence [32]. The episiotomy rate in a study in Ireland, including 323,588 births, varied from between 19% to 27% across four regions, and the rate of AROM varied from between 5% to 9% [33]. A study on variation across 13 regions in Canada, including 8,244 women, found varying episiotomy rates of between 5% to 24% [34]. A variation of between 1% and 34% was found for episiotomy rates in non-instrumental births across many regions in France, in a study including national data [35].

A result in our study which we believe to be significant is the north-south divide within the country. In northern regions, rates of home births were higher and rates of episiotomy in the total population and postpartum oxytocin administration in midwife-led care were lower. It is unlikely that this can be entirely explained by the different risk profiles of low-risk women giving birth at home, compared to low-risk women giving birth in a hospital or a birth centre. Fewer intrapartum referrals to obstetrician-led care might be expected in regions with fewer women in midwife-led care at the onset of labour because women with higher risks may already have been referred earlier during their pregnancy. Yet, our results showed the opposite for two-thirds of all regions. These findings may be explained by regional policies oriented towards obstetrician-led care, or by the preferences of women which may require more referrals. In regions with higher referral rates, PPH rates were higher as well, which may be explained by the association that was found between the augmentation of labour and PPH in our previous publication [23]. Offerhaus et al. (2015) described a similar correlation [15]. Previous studies showed non-urgent referrals, such as meconium-stained amniotic fluid, need for pain medication and/or delay in progress of first stage of labour, being the main reasons for intrapartum referrals in the Netherlands [15, 36]. Our findings of varying rates of intrapartum referral, which may be correlated with higher PPH rates requires further investigation, before any conclusions in terms of policy with regard to augmentation of labour may be drawn.

Several important bodies, such as the WHO, and the series on Caesarean section and Midwifery in the Lancet, have called for action to reduce the inappropriate use of medical interventions in maternity care. Large regional variation that is not explained by differences in maternal characteristics may be unwarranted. There is a lack of literature on regional variations in interventions, related to place of birth and other care processes. It is important to give more insight in the existing variation to care providers and policy makers, to motivate for reflection on their practice policies, on remarkable high or low rates, and on possible causes of variation. This study is therefore an important step in the reduction of unwarranted variation.

**Possible explanations of variations in referrals and interventions.** Many studies have investigated midwives' and obstetricians' perceptions of risk and uncertainty surrounding clinical practice behaviours. They showed that perceived higher risk or uncertainty is associated with higher rates of interventions [37–39] and intrapartum referrals [40]. Care providers' perceptions of risk or uncertainty may also have an impact upon the preferences of childbearing women. Perceptions among care providers may vary due to differences in education—in particular between different countries—underlying social history and various developments in

attitudes towards interventions during childbirth over time and culture [37, 38, 41]. Variations are not merely individual, since care providers are influenced by the culture of their working environment [42]. Similar variations in rates of episiotomy between women in midwife-led care versus those in obstetrician-led care, within the same region, may be explained by a number of factors including: the impact of regional guidelines; comparable attitudes towards interventions during childbirth; and the influence different care providers have upon each other with regard to which care strategies are preferred within a region [37, 38, 42]. Considerable variation between regions with regard to rates of intrapartum referrals, episiotomy and postpartum administration of oxytocin, particularly in midwife-led care, may indicate lack of national consensus about indications for these practices. Further research into the factors which may influence the clinical practice behaviours mentioned above is needed to identify the underlying causes, attitudes towards interventions during childbirth, and the perception of risk [41].

**Episiotomy.**    The routine use of episiotomy is not recommended in recent literature [43] or by the World Health Organization (WHO) [44], because it can lead to physical problems such as a lower pelvic floor muscle strength, dyspareunia and perineal pain [3]. The WHO recommends restricting episiotomy in normal labour to a rate of ten per cent [44], but there are no national guidelines on indications for episiotomy in the Netherlands. An episiotomy rate of 60% for all nulliparous women, found in one of the regions, suggests that this intervention is not performed in a restrictive manner [43]. Since rates of adverse neonatal and maternal outcomes in regions with high episiotomy rates were not lower, such major variation might indicate that episiotomy has been overused in some regions. This warrants further investigation. In our study, a correlation between the regional adjusted ORs for episiotomy, and third or fourth degree perineal tears was not found. In addition, regional variation in performing an episiotomy was considerably larger than the variation in severe perineal tears. An episiotomy is often performed to prevent severe perineal tears [43, 45]. However, literature supports other methods to reduce the rate of third or fourth degree tears which do not have adverse effects [46–48].

**Artificial rupture of membranes.**    AROM may reduce the length of time of labour [4] and it might, therefore, decrease the need for augmentation of labour with oxytocin [4]. On the other hand, some concerns about possible adverse effects have been suggested [4, 49]. The WHO states there should be a valid reason for the artificial rupturing of the membranes in normal labour [44], but there are no national guidelines on indications for AROM in the Netherlands. The average incidence of AROM in our study (47% for nulliparous and 57% for multiparous women in midwife-led care at the onset of labour) was relatively high compared to for instance Germany, where the incidence is 34% and 42% respectively [50]. However, its variation was limited, suggesting a consensus in the use of AROM among midwives. More research is needed to investigate appropriate rates of AROM [4, 44].

**Postpartum administration of oxytocin.**    The large regional variation of postpartum oxytocin among low-risk women, that was found in our study, suggests a lack of a national consensus in midwife-led care. As described by the WHO, the administration of oxytocin seems beneficial for reducing the risk of PPH [44, 51]. Our study suggests underuse of oxytocin in the Netherlands in some regions. This may be explained by some midwives, particularly in regions with low rates, not being convinced of the benefits of routine oxytocin for low-risk women [52], and they may be concerned about potential side effects that have been highlighted in literature [5]. As has been argued before [5], we recommend further research into the use of routine administration of postpartum oxytocin among low-risk women and to develop a national guideline on this issue.

## Conclusions

High rates and large variations were found for intrapartum referral, indicating differences in risk perception between care providers. A correlation was found between intrapartum referral and PPH. More research is required into factors influencing care providers' decision to refer a woman during labour. Higher rates in the use of episiotomy in the total population and post-partum administration of oxytocin in midwife-led care were found in regions with fewer home births. These were not accompanied by better maternal and neonatal outcomes. It seems that existing evidence for restricted use of episiotomy has not been implemented in clinical practice in the Netherlands and large variations in rates of episiotomy and postpartum administration of oxytocin suggest lack of national consensus with regard to these practices. In the short-term, care providers should reflect on their episiotomy practice and restrict the use of episiotomy to evidence-based indications. In the longer-term, policy makers in midwife-led and obstetrician-led care at the national level should achieve consensus on indications for episiotomy and postpartum administration of oxytocin. Further research is needed to identify the reasons for these differences in intervention rates, to explore ways of reducing possibly avoidable interventions, and acquiring evidence-based consensus on the use of interventions.

## Supporting information

**S1 Table. Multivariable logistic regression of process of care and intervention rates by region, adjusted[*] OR [99% CI].** [*]odds ratios, adjusted for parity, maternal age, ethnic background, socioeconomic position and urbanisation.
(DOCX)

**S2 Table. Correlations between process of care variables and interventions.** All correlations are based on adjusted ORs of the intervention rates (adjusted for parity, maternal age, ethnic background, socioeconomic position and urbanisation). A p-value of 0.05 corresponds with a correlation of rho $\geq$ 0.57 or $\leq$ - 0.57 (95% confidence intervals 0.001–0.86). Since the sample size for all measured correlations is the same, namely 12 regions, the correlation is significant at the same value of rho for all measured correlations. Correlations with rho $\geq$ 0.60 or $\leq$ - 0.60 are indicated in bold type since they are considered strong.
(DOCX)

**S3 Table. Correlations between process of care variables or interventions, with obstetric outcomes.** All correlations are based on adjusted ORs of the intervention rates (adjusted for parity, maternal age, ethnic background, socioeconomic position and urbanisation). Correlations for still birth and mortality are not calculated since these were not significantly different between the regions. A p-value of 0.05 corresponds with a correlation of rho $\geq$ 0.57 or $\leq$ - 0.57 (95% confidence intervals 0.001–0.86). Since the sample size for all measured correlations is the same, namely 12 regions, the correlation is significant at the same value of rho for all measured correlations. Correlations with rho $\geq$ 0.60 or $\leq$ - 0.60 are indicated in bold type since they are considered strong.
(DOCX)

## Acknowledgments

We thank Perined for the use of the national database and Janneke Wilschut for giving advice on the statistics.

## Author Contributions

**Conceptualization:** Anna E. Seijmonsbergen-Schermers, Dirkje C. Zondag, Marianne Nieuwenhuijze, Thomas van den Akker, Corine J. Verhoeven, Caroline C. Geerts, François G. Schellevis, Ank de Jonge.

**Formal analysis:** Anna E. Seijmonsbergen-Schermers, Dirkje C. Zondag, Caroline C. Geerts.

**Investigation:** Anna E. Seijmonsbergen-Schermers.

**Methodology:** Anna E. Seijmonsbergen-Schermers, Dirkje C. Zondag, Thomas van den Akker, Caroline C. Geerts.

**Project administration:** Anna E. Seijmonsbergen-Schermers.

**Supervision:** Thomas van den Akker, François G. Schellevis, Ank de Jonge.

**Validation:** Anna E. Seijmonsbergen-Schermers.

**Visualization:** Anna E. Seijmonsbergen-Schermers, François G. Schellevis.

**Writing – original draft:** Anna E. Seijmonsbergen-Schermers.

**Writing – review & editing:** Anna E. Seijmonsbergen-Schermers, Dirkje C. Zondag, Marianne Nieuwenhuijze, Thomas van den Akker, Corine J. Verhoeven, Caroline C. Geerts, François G. Schellevis, Ank de Jonge.

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
