## [Editor Report · Decision Letter 0]

10 Oct 2019

PONE-D-19-18857

Regional variations in childbirth interventions and their correlations with adverse outcomes, birthplace and care provider: a nationwide explorative study

PLOS ONE

Dear Mrs Seijmonsbergen-Schermers,

Thank you for submitting your manuscript to PLOS ONE. After careful consideration, we feel that it has merit but does not fully meet PLOS ONE’s publication criteria as it currently stands. Therefore, we invite you to submit a revised version of the manuscript that addresses the points raised during the review process.

We would appreciate receiving your revised manuscript by Nov 24 2019 11:59PM. To enhance the reproducibility of your results, we recommend that if applicable you deposit your laboratory protocols in protocols.io, where a protocol can be assigned its own identifier (DOI) such that it can be cited independently in the future. For instructions see: http://journals.plos.org/plosone/s/submission-guidelines#loc-laboratory-protocols

We look forward to receiving your revised manuscript.

Kind regards,

Michael Johnson Mahande, PhD

Academic Editor

PLOS ONE

**Journal Requirements:**

**Additional Editor Comments (if provided):**

Section Comment, question, suggestion.

Abstract 1. There was no introduction in the background, the knowledge gap is missing, which could have been reflected in the introduction (i.e. this could have helped to understand why this study was important to be undertaken)

2. No study design is reported on methods

3. Methods not clearly stated

4. Missed the policy implications and recommendations in conclusion.

Introduction

Background 1. Quantify the rates of interventions during childbirth and their variations globally to locally

2. What interventions have been done in Netherlands with respect to childbirth interventions? Are there any Strategic plans? Policies?

Literature review 3. The literature review did not follow pyramid while discussing the previous findings (i.e. provide available evidence about the topic under study from global perspective to local context)

Problem Statement 4. Missed the knowledge gap, what information is lacking/missing in the literature/what we can’t do because we don’t know the trends? (lack of the information to be provided by the present study)

5. What would we benefit from this study findings? (please state it clearly)

Justification 6. Lack of information does not justify a study to be done. (please state clearly the benefits of the study findings; to whom, why and how?)

Objectives 7. Specific objective 2 is not SMART, “We also investigated how these variations are correlated both to each other, and to adverse neonatal and maternal outcomes, adjusted for maternal characteristics.” (this is two in one, it can be separated to make it more specific)

Methodology 1. What was the reason of focusing only on single births?

2. How was confounding controlled for?

3. Individuals from the same facility may have the same characteristics (may be clustered within the facility), how did you account for the clustering effect?

Results 1. Childbirth interventions need to be clearly defined and clearly presented

2. Neonatal and maternal adverse outcomes should be clearly defined

Discussion 1. Missed the comparisons between the study findings and available literature's (similarities & differences)

2. Missed the link of the results and possible explanations basing on what policies, interventions, changes in guidelines were in place at the time.

Strengths and limitations 3. Did the missing values affect study results? if YES what could be the direction of bias?

4. Were there any forms of bias in the study? if YES how these might have influenced the present findings?

5. What is the contribution of the present study to the scientific community?

Conclusion 6. The recommendations were not action oriented (i.e. state clearly, what recommendation, to whom with what action (taking into account the availability of resources): short term and long term action

7. What should be the way forward now that we have the study findings?
---

## [Author Response · Author response to Decision Letter 0]

29 Oct 2019

22 October 2019

Dear dr. Michael Johnson Mahande,

Please find enclosed our revised paper entitled ‘Regional variations in childbirth interventions and their correlations with adverse outcomes, birthplace and care provider: a nationwide explorative study’ (PONE-D-18-17816). We appreciate that we received the opportunity to revise our manuscript. We thank the editors for the constructive comments. We will address the suggestions one by one, referring to the line numbers. In the manuscript, changes are highlighted in red. 

We previously published a related manuscript entitled ‘Regional variations in childbirth interventions in the Netherlands: a nationwide explorative study’ in BMC Pregnancy and Childbirth (doi: 10.1186/s12884-018-1795-0).

When reading the journal requirements, we realized that our Data Availability statement needs to be changed. Our data is available only on request at the Dutch perinatal register “Perined”, and therefore, we are not allowed to provide the data by ourselves. Can you please change the statement into “Data cannot be shared publicly because of restrictions of the perinatal register "Perined". Data are available from Perined (contact via info@perined.nl) for researchers who meet the criteria for access to confidential data, and if Perined gives permission.”

Response to comments

Abstract

1. There was no introduction in the background, the knowledge gap is missing, which could have been reflected in the introduction (i.e. this could have helped to understand why this study was important to be undertaken).

We added the knowledge gap in the abstract. The background section of the abstract is now stating: 

Page 2, lines 25-28:

“Variations in childbirth interventions may indicate inappropriate use. Most variation studies are limited by the lack of adjustments for maternal characteristics and do not investigate variations in adverse outcomes. This study aims to explore regional variations in the Netherlands and their correlations with referral rates, birthplace, interventions and adverse outcomes, adjusted for maternal characteristics.”

2. No study design is reported on methods

We added the following phrase in the methods section of the abstract:

Page 2, line 31:

“In this nationwide retrospective cohort study, using a national data register…”

3. Methods not clearly stated

We added the following phrase in the methods section of the abstract:

Page 2, line 36:

“…with Spearman’s rank correlation coefficients.”

4. Missed the policy implications and recommendations in conclusion.

We added the following sentence in the conclusion section of the abstract:

Page 3, lines 51-52:

“Care providers and policy makers need to be aware of reducing unwarranted variation in birthplace, episiotomy and the postpartum use of oxytocin.”

Introduction/background

5. Quantify the rates of interventions during childbirth and their variations globally to locally.

We added rates of home births, referral, and episiotomy worldwide and in the Netherlands.

Page 4, lines 62-74:

“Worldwide, rates of most interventions and referrals during childbirth have increased [6, 11, 12], episiotomy being the exception [6, 14]. The rate of home births varies worldwide and is low in most high-income countries. For instance, in 2017, the rate of home births was 1% in the USA [15], 0.3% in Australia [16], and 2% in England and Wales [17]. The rate of referrals depends on the maternity care system in a country. Alliman et al. (2016) showed a range of intrapartum referral rates from birth centres to hospitals of between 12% and 37% [18], and Blix et al. (2014) a referral rate range from home to hospital of between 10% and 32% [19]. Episiotomy rates vary largely, from 5% in Denmark, to 75% in Cyprus [20].

In the Netherlands, a variation in intrapartum referral rates among midwifery practices has been shown of between 10% and 64% [21]. Since the late twentieth century, referrals during pregnancy and labour have increased continuously. Where in 1999 more than 60% of women received midwife-led care at the onset of labour, this number has decreased to 51% in 2015 [22, 23]. The rate of home births has historically been high in the Netherlands. However, the rate of home births declined from 23% in 2000 to 13% in 2015 [22, 23]. Episiotomy rates declined from 23% in 2005 to 13% in 2015 in the Netherlands.”

6. What interventions have been done in Netherlands with respect to childbirth interventions? Are there any Strategic plans? Policies?

In the Netherlands, the focus in national policy has been on reducing perinatal mortality. The Steering Committee ‘Pregnancy and Childbirth’ recommended in 2009 more integration in maternity care between midwife-led and obstetrician-led care, which led to several not overarching coordinated initiatives in the field of maternity care. 

The Dutch Organisation of Midwives is aware of the high rate of episiotomy in the Netherlands and has offered intervision sessions for midwives to reflect on their indications for performing an episiotomy in 2017. 

We added the following sentences in box 1 on page 5:

“In 2009, the Steering Committee ‘Pregnancy and Childbirth’ recommended more integration in maternity care between midwife-led and obstetrician-led care, which led to several not overarching coordinated initiatives in the field of maternity care [13].”

Literature review

7. The literature review did not follow pyramid while discussing the previous findings (i.e. provide available evidence about the topic under study from global perspective to local context).

We changed the order in the introduction section, added several sentences, and made changes in the following paragraph.

Page 4, lines 62-74:

“Worldwide, rates of most interventions and referrals during childbirth have increased [6, 11, 12], episiotomy being the exception [6, 14]. The rate of home births varies worldwide and is low in most high-income countries. For instance, in 2017, the rate of home births was 1% in the USA [15], 0.3% in Australia [16], and 2% in England and Wales [17]. The rate of referrals depends on the maternity care system in a country. Alliman et al. (2016) showed a range of intrapartum referral rates from birth centres to hospitals of between 12% and 37% [18], and Blix et al. (2014) a referral rate range from home to hospital of between 10% and 32% [19]. Episiotomy rates vary largely, from 5% in Denmark, to 75% in Cyprus [20].

In the Netherlands, a variation in intrapartum referral rates among midwifery practices has been shown of between 10% and 64% [21]. Since the late twentieth century, referrals during pregnancy and labour have increased continuously. Where in 1999 more than 60% of women received midwife-led care at the onset of labour, this number has decreased to 51% in 2015 [22, 23]. The rate of home births has historically been high in the Netherlands. However, the rate of home births declined from 23% in 2000 to 13% in 2015 [22, 23]. Episiotomy rates declined from 23% in 2005 to 13% in 2015 in the Netherlands.”

Problem statement

8. Missed the knowledge gap, what information is lacking/missing in the literature/what we can’t do because we don’t know the trends? (lack of the information to be provided by the present study)

9. What would we benefit from this study findings? (please state it clearly).

We believe the following sentence of our introduction section clearly explains the knowledge gap. We added a sentence on the benefits from our study findings.

Page 6, lines 100-102:

“Little information is available as to how regional variations in rates of referral, place of birth, and interventions during childbirth, relate to each other, nor how they might relate to adverse neonatal and maternal outcomes.”

Page 6, lines 102-106:

“Knowledge on these correlation will give insight into underlying processes of variations in childbirth interventions, place of birth, and referral, and will help care providers and policy makers to know which variation is large and likely unwarranted and should therefore be the focus of changes in practices and policies with the ultimate aim to improve the quality of maternity care.”

Justification

10. Lack of information does not justify a study to be done. (please state clearly the benefits of the study findings; to whom, why and how?)

See the response to comments 8 and 9.

Objectives

11. Specific objective 2 is not SMART, “We also investigated how these variations are correlated both to each other, and to adverse neonatal and maternal outcomes, adjusted for maternal characteristics.” (this is two in one, it can be separated to make it more specific).

We separated our research aim into three aims.

Page 6, lines 107-112:

“The first aim of this study was to explore which regional variations in intrapartum rates of referral, place of birth, and use of intrapartum interventions, exist for women who gave birth in the Netherlands between 2010 and 2013. Secondly, we aimed to investigate how these variations are correlated to each other. Thirdly, we examined the association between variations, and adverse neonatal and maternal outcomes, adjusted for maternal characteristics.”

Methodology

12. What was the reason of focusing only on single births? 

We focused on single births because multiple pregnancies are associated with much higher risks of adverse outcomes and therefore medical interventions are often justified. Most unwarranted variation in interventions occur among women at low risk of complications. Therefore, we only focused on single births. 

Page 6, line 117-119:

“We focused on single births because multiple pregnancies are associated with much higher risks of adverse outcomes and therefore medical interventions are often justified.”

13. How was confounding controlled for?

The analyses were conducted on the level of the 614,730 women. The region was the independent variable in the analyses, and the primary or secondary outcome was the dependent variable. In the multivariable analyses, we adjusted for baseline characteristics, because these could differ across the regions and could therefore have been an explanation of the variation.

We added some sentences in the data analyses section to make clearer how our analyses were conducted. 

Page 8, lines 157-166:

“The analyses were conducted on the level of the women, with the region as independent variable, and the primary or secondary outcome as dependent variable. In the logistic regression analyses, the overall rate of the outcome we investigated, weighted for the number of women per region, was considered as a reference. Univariable analyses were conducted in order to gain insight into the variations in the rates of primary and secondary outcomes among the twelve regions. These were followed by multivariable logistic regression analyses with adjustments for: maternal age (<40 years, 40 years or older); ethnic background (Dutch, non-Dutch); socioeconomic position (low, medium or high) - based on postal code and education, employment and level of income; and the degree of urbanisation (rural, intermediate or urban) - based on women’s residential postal code (Statistics Netherlands).”

14. Individuals from the same facility may have the same characteristics (may be clustered within the facility), how did you account for the clustering effect?

We had the following considerations for not performing multilevel analyses to adjust for clustering.

The aim of our study is to describe and explore which variations exist and how variations in interventions or processes relate to each other and to adverse outcomes. These variations might (partly) be explained by clustering of women in care facilities (birth centre, hospital, midwifery practice) and care providers. Variations in clinical practice that are not explained by differences in obstetric risks or preferences of women, may be unwarranted. If we would conduct multilevel analyses, we would adjust for the differences between care providers or other levels. However, these variations are exactly the topic of our study and we do not want control for them. Future research should use multilevel analyses to investigate whether differences between care facilities and care providers may explain the variations between regions, but that is beyond the scope of our study. 

Results

15. Childbirth interventions need to be clearly defined and clearly presented.

16. Neonatal and maternal adverse outcomes should be clearly defined.

We added three subtitles in the manuscript under the subheading ‘Selection of variables’:

- Care processes

- Interventions

- Adverse outcomes

The outcome measures were defined as follows:

Page 7-8 , lines 131-151:

“Care processes

The following primary outcome variables concerning maternity care processes were examined: the number of women receiving midwife-led care at the onset of their pregnancy, at the onset of labour, and at the time of birth; intrapartum referral to obstetrician-led care, and the planned, and actual place of birth (home, hospital and birth centre midwife-led, hospital obstetrician-led). The onset of labour was defined as the onset of uterine contractions or rupture of membranes. Intrapartum referral to an obstetrician-led care setting was defined as a referral after the onset of labour and before birth. The planned place of birth was defined as that at the onset of labour. Therefore, women who were referred, during pregnancy and before labour, to obstetrician-led care could not have a birth planned in midwife-led care. 

Interventions

The following primary outcome variables concerning interventions were examined: the rates of episiotomy in vaginal births; AROM, and postpartum administration of oxytocin. Data about AROM and postpartum administration of oxytocin were only available in the midwifery part of the perinatal database and were therefore not described for the obstetrician-led care group. 

Adverse outcomes

Secondary neonatal and maternal outcomes were: antepartum and intrapartum stillbirth; neonatal mortality up to seven days; Apgar score below seven at five minutes; third or fourth degree perineal tear among vaginal births; and postpartum haemorrhages (PPH) of more than 1,000 ml.”

Discussion

17. Missed the comparisons between the study findings and available literatures (similarities & differences).

We added a paragraph on previous literature on regional variations in the discussion section. 

Pages 18-19, lines 317-327:

“Previous studies on regional variations in other countries, showed a variety in intervention rates across studies. When focusing on the subjects of our study, we found lower rates and less variation in rates of AROM, and, similar to our results, large variations in episiotomy rates. Studies were found on regional variations in episiotomy and AROM in Brazil, Ireland, Canada, and France, but there is a lack of literature on regional variations in care processes. A Brazilian study, including 23,940 women, found rates of AROM varying from between 32% to 48%, and episiotomy rates varying from 49% to 69% across five regions, although variation in episiotomy rates was only significantly different in the region with the highest incidence [32]. The episiotomy rate in a study in Ireland, including 323,588 births, varied from between 19% to 27% across four regions, and the rate of AROM varied from between 5% to 9% [33]. A study on variation across 13 regions in Canada, including 8,244 women, found varying episiotomy rates of between 5% to 24% [34]. A variation of between 1% and 34% was found for episiotomy rates in non-instrumental births across many regions in France, in a study including national data [35].” 

18. Missed the link of the results and possible explanations basing on what policies, interventions, changes in guidelines were in place at the time.

Criteria for referral from midwife-led to obstetrician-led care have been laid out in the obstetric indication list of 2003. 

Page 5, box 1:

“Criteria for referral from midwife-led to obstetrician-led care have been laid out in the obstetric indication list of 2003.“

However, it is possible that regional protocols differ. In the following (added) parts of the discussion section we suggest that differences in policies and culture may lead to variations in interventions.

Page 19, lines 338-341:

“Yet, our results showed the opposite for two-thirds of all regions. These findings may be explained by regional policies oriented towards obstetrician-led care, or by the preferences of women which may require more referrals.”

Page 20, lines 366-373:

“Similar variations in rates of episiotomy between women in midwife-led care versus those in obstetrician-led care, within the same region, may be explained by a number of factors including: the impact of regional guidelines; comparable attitudes towards interventions during childbirth; and the influence different care providers have upon each other with regard to which care strategies are preferred within a region [37, 38, 42]. Considerable variation between regions with regard to rates of intrapartum referrals, episiotomy and postpartum administration of oxytocin, particularly in midwife-led care, may indicate lack of national consensus about indications for these practices.”

Page 20, lines 380-382:

“The WHO recommends restricting episiotomy in normal labour to a rate of ten per cent [44], but there are no national guidelines on indications for episiotomy in the Netherlands.”

Page 21, lines 395-397

“The WHO states there should be a valid reason for the artificial rupturing of the membranes in normal labour [44], but there are no national guidelines on indications for AROM in the Netherlands.”

Page 22, lines 410-411:

“…we recommend further research into the use of routine administration of postpartum oxytocin among low-risk women and to develop a national guideline on this issue.”

Strengths and limitations

19. Did the missing values affect study results? if YES what could be the direction of bias?

Missing values were very few in number, and therefore we do not think that they influenced the results. We changed this as follows:

Page 16-17, line 300-302:

“The number of missing values was very low; there were no outcome variables with more than 1.5% of values missing, and no characteristic variables with more than 2.5% of values missing.”

20. Were there any forms of bias in the study? if YES how these might have influenced the present findings?

We think the most important form of bias is reporting bias, which we discussed in the limitations and strengths section. Because our independent variable of interest is the region, and we do no have any indication that this reporting bias might vary across regions, we do not think that this has influenced the findings. 

21. What is the contribution of the present study to the scientific community?

Several important bodies, such as the WHO, and the series on Caesarean section and Midwifery in the Lancet, have called for action to reduce the inappropriate use of medical interventions in maternity care. Large regional variation that is not explained by differences in maternal characteristics may be unwarranted. There is a lack of literature on regional variations in interventions, related to place of birth and other care processes. It is important to give more insight in the existing variation to care providers and policy makers, to motivate for reflection on their practice policies, on remarkable high or low rates, and on possible causes of variation. This is an important step in the reduction of unwarranted variation.

Page 17, lines 279-280:

“This study is a first step towards understanding the appropriate use of interventions in childbirth, as was recommended in a Lancet series on Midwifery [7].”

Page 20, lines 349-356:

“Several important bodies, such as the WHO, and the series on Caesarean section and Midwifery in the Lancet, have called for action to reduce the inappropriate use of medical interventions in maternity care. Large regional variation that is not explained by differences in maternal characteristics may be unwarranted. There is a lack of literature on regional variations in interventions, related to place of birth and other care processes. It is important to give more insight in the existing variation to care providers and policy makers, to motivate for reflection on their practice policies, on remarkable high or low rates, and on possible causes of variation. This study is therefore an important step in the reduction of unwarranted variation.”

Conclusion

22. The recommendations were not action oriented (i.e. state clearly, what recommendation, to whom with what action (taking into account the availability of resources): short term and long term action.

23. What should be the way forward now that we have the study findings?

The following sentences were added to the conclusion section:

Page 22, lines 419-422:

“In the short-term, care providers should reflect on their episiotomy practice and restrict the use of episiotomy to evidence-based indications. In regions with high episiotomy rates, care providers should focus on decreasing this rate. In the longer-term, policy makers in midwife-led and obstetrician-led care at national level should achieve consensus, on indications for episiotomy and postpartum administration of oxytocin.”

We are confident that our study is a first, important step in reducing the inappropriate use of medical interventions in maternal and newborn care, which is called for by important bodies, such as the WHO. We therefore hope that you will consider publication of our article. 

Yours faithfully, on behalf of all authors,

Anna Seijmonsbergen-Schermers, MSc

---

## [Decision Letter · Decision Letter 1]

3 Feb 2020

PONE-D-19-18857R1

Regional variations in childbirth interventions and their correlations with adverse outcomes, birthplace and care provider: a nationwide explorative study

PLOS ONE

Dear Mrs Seijmonsbergen-Schermers,

Thank you for submitting your manuscript to PLOS ONE. After careful consideration, we feel that it has merit but does not fully meet PLOS ONE’s publication criteria as it currently stands. Therefore, we invite you to submit a revised version of the manuscript that addresses the points raised during the review process.

We would appreciate receiving your revised manuscript by Mar 19 2020 11:59PM. To enhance the reproducibility of your results, we recommend that if applicable you deposit your laboratory protocols in protocols.io, where a protocol can be assigned its own identifier (DOI) such that it can be cited independently in the future. For instructions see: http://journals.plos.org/plosone/s/submission-guidelines#loc-laboratory-protocols

We look forward to receiving your revised manuscript.

Kind regards,

Michael Johnson Mahande, PhD

Academic Editor

PLOS ONE

Reviewers' comments:

Reviewer's Responses to Questions

**Comments to the Author**

1. If the authors have adequately addressed your comments raised in a previous round of review and you feel that this manuscript is now acceptable for publication, you may indicate that here to bypass the “Comments to the Author” section, enter your conflict of interest statement in the “Confidential to Editor” section, and submit your "Accept" recommendation.

Reviewer #1: All comments have been addressed

Reviewer #2: (No Response)

2. Is the manuscript technically sound, and do the data support the conclusions?

Reviewer #1: Yes

Reviewer #2: Yes

3. Has the statistical analysis been performed appropriately and rigorously? 

Reviewer #1: Yes

Reviewer #2: Yes

4. Have the authors made all data underlying the findings in their manuscript fully available?

Reviewer #1: Yes

Reviewer #2: Yes

5. Is the manuscript presented in an intelligible fashion and written in standard English?

Reviewer #1: Yes

Reviewer #2: Yes

6. Review Comments to the Author

Reviewer #1: this is an interesting paper addressing regional variations in obstetrical care in The Netherlands. The paper has improved based on reviewers comments that were adequately addressed.

Reviewer #2: Re: Regional variations in childbirth interventions and their correlations with adverse outcomes, birthplace and care provider: a nationwide explorative study

I have read this manuscript within an important topic, rates of referrals and interventions, with interest. The manuscript includes a lot of information about regions, referral rates, intervention rates, and maternal and neonatal outcomes. This amount of information requires a good organisation in the paper and I think the paper is quite well organised and give valuable information. I have some comments listed below:

Introduction

• I cannot find references number 22 and 23. Please add web address or other publication details.

Methods

• I would recommend using “RS” for the Spearman rank correlation coefficient. Use of “p-value” confuses readers, I think.

Results

• Again, confusing symbol used for the correlation coefficient.

• Intrapartum referrals from midwife-led to obstetrician care seem very high, 55% to 68% for nulliparous women. Is this correct, are these referrals done during labour and birth? During active labour?

• There is a misspelling in the title for figure 4 (aArtificial).

• In the tables, GR, DR, ZL etc need explanation, I suggest under the table.

Discussion

• These results give an important message – wide variations in intervention rates depend to a large extent on different cultures in childbirth care – but I miss this message being highlighted in the conclusion. I would like to see the authors’ conclusions on referral rates.

• I am a little confused about the choice of subheadings in the Discussion section. First a summary, good, and the strengths and limitation, good. But what is meant by “Literature”? Is that a subheading or a sub-subheading after “Interpretation…”? I think this could be made clearer.

• Subheading “Possible explanation….” seems to be about referral rates – maybe include “referral rates” in the subheading would make it easier to find those discussed. Thereafter the interventions studied have separate sub-headings so why not referrals?

• I understand that it is a little complicated to get all studied factors into the conclusion but I think it is worth a try.

7. PLOS authors have the option to publish the peer review history of their article (what does this mean?). If published, this will include your full peer review and any attached files.

Reviewer #1: Yes: Marleen Temmerman

Reviewer #2: Yes: Anna Dencker

---

## [Author Response · Author response to Decision Letter 1]

6 Feb 2020

6 February 2020

Dear dr. Michael Johnson Mahande,

Please find enclosed our revised paper entitled ‘Regional variations in childbirth interventions and their correlations with adverse outcomes, birthplace and care provider: a nationwide explorative study’ (PONE-D-19-18857R1). We appreciate that we received the opportunity to revise our manuscript. We thank the reviewer for the constructive comments. We will address the suggestions one by one, referring to the line numbers. In the manuscript, changes are highlighted in red. 

Response to comments of Reviewer 2

Introduction

- I cannot find references number 22 and 23. Please add web address or other publication details.

We added the web address of these two references.

Page 28, lines:

“22. Perined. Perinatale Zorg in Nederland 2015 Utrecht: Perined; 2016 [Accessed 03-Feb-2020]. Available from: https://assets.perined.nl/docs/980021f9-6364-4dc1-9147-d976d6f4af8c.pdf.

23. Stichting Perinatale Registratie Nederland. Perinatale Zorg in Nederland 2005 Utrecht: Stichting Perinatale Registratie Nederland; 2008 [Accessed 03-Feb-2020]. Available from: https://assets.perined.nl/docs/2f0a701a-d8f3-4cbd-98a1-2c4e588cf435.pdf.”

Methods

- I would recommend using “RS” for the Spearman rank correlation coefficient. Use of “p-value” confuses readers, I think.

We changed all “ρ-signs” into “rho” in the manuscript as well as in the table. We highlighted these changes in the manuscript.

Results

- Again, confusing symbol used for the correlation coefficient.

See our previous response.

- Intrapartum referrals from midwife-led to obstetrician care seem very high, 55% to 68% for nulliparous women. Is this correct, are these referrals done during labour and birth? During active labour?

These percentages are indeed correct. It concerns referrals between the start of labour and birth. The start of labour is defined as active uterine contractions or rupture of the membranes. The definition of intrapartum referral was described on page 8, lines 136-137 and the definition of onset of labour on lines 135-136:

“The onset of labour was defined as the onset of active uterine contractions or rupture of the membranes. Intrapartum referral to an obstetrician-led care setting was defined as a referral after the onset of labour and before birth.”

- There is a misspelling in the title for figure 4 (aArtificial).

We changed this accordingly. 

Page 12, line 240:

“Fig 7. Interregional variation in Artificial rupture of membranes in midwife-led care at onset of labour”

- In the tables, GR, DR, ZL etc need explanation, I suggest under the table.

We added an explanation under the table.

Page 15, lines 247-248:

“GR=Groningen; FR=Friesland; DR=Drenthe; OV=Overijssel; FL=Flevoland; GD=Gelderland; UT=Utrecht; NH=Noord-Holland; ZH=Zuid-Holland; NB=Noord-Brabant; LB=Limburg. These regions are the twelve provinces in the Netherlands.”

Discussion

- These results give an important message – wide variations in intervention rates depend to a large extent on different cultures in childbirth care – but I miss this message being highlighted in the conclusion. I would like to see the authors’ conclusions on referral rates.

We added the following sentences into the conclusions:

Page 22, lines 416-419:

“High rates and large variations were found for intrapartum referral, indicating differences in risk perception and possibly in childbirth care culture between care providers. A correlation was found between intrapartum referral and PPH. More research is required into factors influencing care providers’ decision to refer a woman during labour.”

- I am a little confused about the choice of subheadings in the Discussion section. First a summary, good, and the strengths and limitation, good. But what is meant by “Literature”? Is that a subheading or a sub-subheading after “Interpretation…”? I think this could be made clearer.

We changed this sub-subheading into “Interpretation in view of previous literature” on page 18, line 321.

- Subheading “Possible explanation….” seems to be about referral rates – maybe include “referral rates” in the subheading would make it easier to find those discussed. Thereafter the interventions studied have separate sub-headings so why not referrals?

We changed this heading into “Possible explanations of variations in referrals and interventions” on page 20, line 360.

- I understand that it is a little complicated to get all studied factors into the conclusion but I think it is worth a try.

We added the following sentences into the conclusions:

Page 22, lines 416-419:

“High rates and large variations were found for intrapartum referral, indicating differences in risk perception and possibly in childbirth care culture between care providers. A correlation was found between intrapartum referral and PPH. More research is required into factors influencing care providers’ decision to refer a woman during labour.”

We are confident that our study is a first, important step in reducing the inappropriate use of medical interventions in maternal and newborn care, which is called for by important bodies, such as the WHO. We therefore hope that you will consider publication of our article. 

Yours faithfully, on behalf of all authors,

Anna Seijmonsbergen-Schermers, MSc

---

## [Editor Report · Decision Letter 2]

10 Feb 2020

Regional variations in childbirth interventions and their correlations with adverse outcomes, birthplace and care provider: a nationwide explorative study

PONE-D-19-18857R2

Dear Dr. Seijmonsbergen-Schermers,

We are pleased to inform you that your manuscript has been judged scientifically suitable for publication and will be formally accepted for publication once it complies with all outstanding technical requirements.

With kind regards,

Michael Johnson Mahande, PhD

Academic Editor

PLOS ONE
---

## [Editor Report · Acceptance letter]

13 Feb 2020

PONE-D-19-18857R2 

Regional variations in childbirth interventions and their correlations with adverse outcomes, birthplace and care provider: a nationwide explorative study 

Dear Dr. Seijmonsbergen-Schermers:

I am pleased to inform you that your manuscript has been deemed suitable for publication in PLOS ONE. Congratulations! Your manuscript is now with our production department. 

With kind regards,

on behalf of

Dr. Michael Johnson Mahande 

Academic Editor

PLOS ONE